# Associations between future health expectations and patient satisfaction after lumbar spine surgery: a longitudinal observational study of 9929 lumbar spine surgery procedures

Anders Joelson [1,2], Lilla Szigethy,[2] Peter Wildeman,[1,2] Freyr Gauti Sigmundsson,[1,2] Jan Karlsson[3]

¹School of Medical Sciences, Orebro University, Orebro, Sweden
²Department of Orthopaedics, Orebro University Hospital, Orebro, Sweden
³Faculty of Medicine and Health, Orebro University, Orebro, Sweden

**Correspondence to**
Dr Anders Joelson;
anders@joelson.se

## ABSTRACT

**Objective** This study aimed to investigate the associations between general health expectations and patient satisfaction with treatment for the two common spine surgery procedures diskectomy for lumbar disk herniation (LDH) and decompression for lumbar spinal stenosis (LSS).

**Design** Register study with prospectively collected preoperative and 1-year postoperative data.

**Setting** National outcome data from Swespine, the national Swedish spine register.

**Participants** A total of 9929 patients, aged between 20 and 85 years, who were self-reported non-smokers, and were operated between 2007 and 2016 for one-level LSS without degenerative spondylolisthesis, or one-level LDH, were identified in the national Swedish spine register (Swespine). We used SF-36 items 11c and 11d to assess future health expectations and present health perceptions. Satisfaction with treatment was assessed using the Swespine satisfaction item.

**Interventions** One-level diskectomy for LDH or one-level decompression for LSS.

**Primary outcome measures** Satisfaction with treatment.

**Results** For LSS, the year 1 satisfaction ratio among patients with negative future health expectations preoperatively was 60% (95% CI 58% to 63%), while it was 75% (95% CI 73% to 76%) for patients with positive future health expectations preoperatively. The corresponding numbers for LDH were 73% (95% CI 71% to 75%) and 84% (95% CI 83% to 85%), respectively.

**Conclusions** Patients operated for the common lumbar spine diseases LSS or LDH, with negative future general health expectations, were significantly less satisfied with treatment than patients with positive expectations with regard to future general health. These findings are important for patients, and for the surgeons who counsel them, when surgery is a treatment option for LSS or LDH.

## INTRODUCTION

Lumbar degenerative spine diseases are major causes of pain and disability worldwide.[1 2] Patient-reported outcome measures (PROMs)

---

## STRENGTHS AND LIMITATIONS OF THIS STUDY

⇒ The study includes a large number of patients from a national database with high, stable coverage.
⇒ We recognise the inherent limitations of register data, such as lack of confounder information, missing data or unknown data quality.
⇒ The data were incomplete in 58% of the procedures, which is a major limitation of our study that affects the internal and external validity of our findings.

---

and patient satisfaction scales are commonly used to evaluate treatment outcomes after lumbar spine surgery.[3] However, there are inconsistencies between PROM changes and treatment satisfaction when evaluating surgical outcome. For example, Chotai *et al*[4] found that 83% of patients were satisfied with treatment after elective surgery for degenerative spine disease, whereas only 62% achieved minimal important change for the Oswestry/ neck disability indices. Furthermore, Godil *et al*,[5] in an analysis using receiver operating characteristics curves, found that improvement in the Oswestry disability index failed to discriminate between satisfaction and dissatisfaction with treatment with good accuracy after spine surgery. In contrast, Copay *et al*[6] found a strong association between the Oswestry disability index and patient satisfaction after lumbar surgery. The variety of results suggest that patient satisfaction is also influenced by factors other than PROM changes, such as expectations, socioeconomic factors and mental health.

The impact of psychological factors on the outcomes of spine surgery has been thoroughly researched.[7–10] In addition, Iderberg *et al*[11] demonstrated that socioeconomic indicators are associated with the outcomes

**Table 1** Questions and response options

| | Question | Response options |
|---|---|---|
| Future health (SF-36 item 11c) | I expect my health to get worse. | 1. Definitely true<br>2. Mostly true<br>3. Don't know<br>4. Mostly false<br>5. Definitely false |
| Present health (SF-36 item 11d) | My health is excellent. | 1. Definitely true<br>2. Mostly true<br>3. Don't know<br>4. Mostly false<br>5. Definitely false |
| Satisfaction (Swespine) | What is your attitude regarding the outcome of your spine surgery? | 1. I am satisfied<br>2. I am uncertain<br>3. I am dissatisfied |

of surgery for lumbar spinal stenosis. Moreover, several reports have shown that preoperative expectations on recovery predict the outcome of spine surgery.[12–19] Interestingly, previous studies have shown that expectations on future general health are associated with mortality and functional decline.[20–22] These findings raise the question of whether there is also an association between future general health expectations and the outcome of health interventions like spine surgery. However, data on the expectations on future general health expectations and outcomes of spine surgery are limited. Therefore, the current study aimed to investigate the associations between future general health expectations and patient satisfaction following surgery for degenerative spine diseases.

## METHODS
### Study design
The present study was a register study, with prospectively collected longitudinal data from Swespine, the national Swedish spine register.[23]

### The national Swedish spine register (Swespine)
The Swespine register was launched in 1992 and covers 90% of the spine units in Sweden. The 1-year follow-up rate is 70–75%.[23] The register includes data on diagnoses,

surgical procedures, complications and PROMs. The surgeon is responsible for submitting data about the surgery.

### Patient data set
Patients, aged between 20 and 85 years, who were self-reported non-smokers, and were surgically treated between 2007 and 2016 for one-level lumbar spinal stenosis (LSS) without degenerative spondylolisthesis, or one-level lumbar disk herniation (LDH), were identified in Swespine.

### Measures
The SF-36 is an 8-dimensional, 36-item, self-administered health-related quality of life (HRQoL) instrument for the assessment of general HRQoL.[24] The instrument has six items for assessment of general health perceptions: item 1 (present health), item 2 (health transition), item 11a (health comparison), item 11b (health context), item 11c (future health) and item 11d (present health). Items 1 and 11a–d form the general health domain of SF-36. In our study, we used items 11c and 11d to assess future health expectations and present health perceptions (table 1). We grouped future health expectations into negative (pessimistic) health expectations (item 11c response options 1, 2 and 3) and positive (optimistic) health expectations (item 11c response options 4 and 5). We grouped present health perceptions into positive present health perceptions (item 11d response options 1, 2 and 3) and negative present health perceptions (item 11c response options 4 and 5). We used the Swedish translation of SF-36 V.1 in our study.[25]

Satisfaction with treatment was assessed using the Swespine satisfaction item (table 1). In our analysis, we grouped satisfaction with treatment into satisfied (response option 1) and dissatisfied (response options 2 and 3).

### Statistics
Data are presented as mean and SD and/or 95% CIs. Bootstrapping was used to calculate the CIs.[26] Standardised response mean (SRM) for paired data was used to evaluate effect size.[27] The SRM was interpreted as follows: <0.2 no effect, 0.2–0.4 small effect, 0.5–0.7 moderate effect, >0.7 large effect.[28] Multiple linear logistic regression analysis

**Table 2** Comparison of characteristics between patients with positive and negative future health expectations preoperatively

| | LSS (n=3969) | | LDH (n=5960) | |
|---|---|---|---|---|
| | Negative future health expectations | Positive future health expectations | Negative future health expectations | Positive future health expectations |
| n (%) | 1501 (37.8) | 2468 (62.2) | 1333 (22.4) | 4627 (77.6) |
| Age, mean (SD) | 67.8 (10) | 64.9 (10.1) | 46.8 (13.8) | 44.3 (12.1) |
| BMI, mean (SD) | 27.7 (4.0) | 27.5 (4.0) | 26.7 (4.4) | 26.1 (4.0) |
| Women, n (%) | 709 (47.2) | 1110 (45) | 596 (44.7) | 2039 (44.1) |

BMI, body mass index; LDH, lumbar disk herniation; LSS, lumbar spinal stenosis.

**Table 3** Patient satisfaction 1 year postoperatively for LSS (n=3969) and LDH (n=5960)

| | Negative future health expectations preoperatively | Positive future health expectations preoperatively |
|---|---|---|
| LSS, % satisfied (95% CI) (n/total) | 60.4 (57.8; 62.8) (906/1501) | 74.5 (72.9; 76.1) (1839/2468) |
| LDH, % satisfied (95% CI) (n/total) | 73 (70.7; 75.2) (973/1333) | 83.8 (82.8; 84.9) (3879/4627) |

LDH, lumbar disk herniation; LSS, lumbar spinal stenosis.

was used to model the relationship between the outcome and covariates.[29] All covariates of the model were binary; continuous covariates were dichotomised by using their respective median values.

## Patient and public involvement

The patients and the public were not involved in the design, recruitment, conduct or dissemination plans of this research.

## RESULTS

A total of 24 127 surgical procedures for the treatment of the lumbar spine diseases LDH and LSS were included in Swespine between 2007 and 2016. Preoperative or 1-year postoperative SF-36 data were incomplete for 14 198 (58%) of the procedures which provided 9929 procedures eligible for analysis. The baseline characteristics of the included and excluded patients are presented in online supplemental table 1.

For LSS, 1501 (38%) of 3969 patients had negative future health expectations preoperatively and 2117 (53%) of 3969 patients had negative future health expectations at the year 1 follow-up. For LDH, the corresponding number was 1333 (22%) of 5960 patients preoperatively and 2047 (34%) of 5960 patients at the year 1 follow-up (online supplemental table 2).

The preoperative characteristics of the patients with negative and positive future health expectations are presented in table 2. The SF-36 profiles preoperatively and 1 year postoperatively are shown in online supplemental figures 1 and 2, and the effect sizes of changes are shown in online supplemental tables 3 and 4. For LSS, the satisfaction ratio year 1 postoperatively among patients with negative future health expectations preoperatively was 60%, while it was 75% for patients with positive future health expectations (table 3). The corresponding levels for LDH were 73% and 84%, respectively. The differences in satisfaction ratios were statistically significant (non-overlapping CIs). Table 4 summarises multiple linear logistic regression models for patient satisfaction 1 year postoperatively using preoperative future health expectations, preoperative present health perceptions, age, gender and body mass index (BMI) as covariates.

## DISCUSSION

In this paper, we found that patients, operated for the common lumbar spine diseases LSS or LDH, with negative future general health expectations preoperatively, were significantly less satisfied with treatment compared with patients with positive expectations with regard to future general health. To our knowledge, our study is the first to report data on the association between expectations on future general health assessed preoperatively and patient satisfaction after lumbar spine surgery. Clinicians that are using SF-36 in the preoperative evaluation of patients scheduled for LSS or LDH surgery will get additional information by analysing the answer to item 11c. Assessment of item 11c might be useful and valuable in practice settings in the identification of patients who might benefit from a more active rehabilitation or follow-up.

Belayneh et al[19] studied the impact of future health expectations on the outcome after surgical repair of proximal humeral fractures and found that patients with positive expectations on their health, early following the injury, had better long-term outcomes. The authors evaluated future health expectations using a question with exactly the same wording as used in our study. This strengthens the assumption that SF-36 item 11c may be used to assess future health expectations in the field of orthopaedic surgery. We agree with the authors that healthcare providers should communicate with patients, to ensure that they are setting clear expectations of the benefits and risks for each patient.

Iversen et al[12] studied several expectations summed across the domains pain reduction, physical functioning and social functioning to evaluate the prognostic importance of preoperative expectations on the treatment

**Table 4** Multiple linear logistic regression models for patient satisfaction 1 year postoperatively for LSS (n=3969) and LDH (n=5960)

| | LSS | LDH |
|---|---|---|
| Intercept, OR (95% CI) | 0.981 (0.834; 1.15) | 1.9 (1.6; 2.25) |
| Future health, OR (95% CI) | 1.71 (1.48; 1.97) | 1.72 (1.48; 1.99) |
| Present health, OR (95% CI) | 1.42 (1.23; 1.64) | 1.47 (1.28; 1.69) |
| Age, OR (95% CI) | 1.38 (1.2; 1.59) | 1.14 (0.994; 1.3) |
| Gender, OR (95% CI) | 1.13 (0.98; 1.29) | 1.17 (1.02; 1.33) |
| BMI, OR (95% CI) | 1.25 (1.09; 1.44) | 1.14 (0.998; 1.3) |

Hosmer-Lemeshow goodness-of-fit test: LSS p=0.52, LDH p=0.48. BMI, body mass index; LDH, lumbar disk herniation; LSS, lumbar spinal stenosis.

outcomes of LSS surgery and found that patients' expectations influence recovery from surgery at 6 months. The authors concluded that clinicians should discuss expectations with patients preoperatively in order to establish realistic goals and to enable patients to actively engage in their rehabilitation, a conclusion that we agree with.

Standard surgical procedures for the treatment of LDH and LSS are considered safe and beneficial treatment options.[30 31] However, there are rare but serious complications such as nerve rot lesions.[31] As the main goal of elective surgery for LDH and LSS is to improve patient quality of life, it is important to weigh the benefits against the potential risks when discussing treatment options with patients. For patients with negative future health expectations, our data suggest that the satisfaction rate for LSS surgery could be as low as 60%. This information is important from a shared decision-making perspective when balancing the benefits and risks of surgery.

We used SF-36 item 11c to assess future health expectations. Previous studies have indicated that the wording of item 11c sometimes is seen to be unnecessarily negative.[32] Furthermore, Sharples et al[33] speculated that elderly people might be reluctant to consider questions about worsening in health but concluded that the item did not affect the internal consistency of the SF-36 general health domain. Although there are some concerns about the design of item 11c, we do not expect that these concerns would invalidate the use of item 11c for the assessment of future health expectations.

Several factors may affect future health expectations. The SF-36 health profiles presented in online supplemental figures 1 and 2 and online supplemental tables 3 and 4 indicate that negative future health expectations do not only affect the general health domain, as the patients report lower scores on all SF-36 domains. This illustrates that negative future health expectations affect several dimensions of HRQoL.

The five-factor model is commonly used in psychology to model different personality traits.[34] The model uses five orthogonal trait dimensions to describe different personalities: neuroticism (N), extraversion (E), openness (O), agreeableness (A) and conscientiousness (C). Chapman et al[35] studied the influence of the five-factor model personality traits on perceived health using the NEO Five-Factor Inventory[36] and SF-36. Low N scores and high E scores were associated with positive future health expectations. However, although the differences in the personality trait scores were statistically significant, the actual differences were small. Hendriks et al[37] found that patient satisfaction was only marginally associated with personality. Consequently, the influence of different personality traits on health expectations and patient satisfaction remains unclear, and this field may benefit from further research.

Notably, 38% of the patients with LSS had negative future health expectations preoperatively, whereas 53% had negative future health expectations at the year 1 follow-up. The corresponding levels for LDH were 22%

and 34%, respectively. One possible explanation is that preoperatively, the patients expect an improvement in health because of the forthcoming operation, while at 1 year after the operation, the patients may be more neutral or pessimistic about future health improvements. This indicates that questions about future health expectations must be interpreted with caution when asked before and after a health intervention.

The results of our multiple linear logistic regression analysis indicated that there was an association between general health assessments (present and future) and patient satisfaction after surgery for LDH and LSS. Ferraro and Wilkinson[21] reported that there are indications that queries about future health expectations are more useful than those about past health changes in mortality predictions. However, our study could not confirm that future health expectations had larger impact on patient satisfaction than present health perceptions since the ORs for present and future health had overlapping CIs. Age, gender and BMI made only minor contributions as predictors of patient satisfaction 1 year after LDH surgery, whereas age had some impact on satisfaction after surgery for LSS.

Our findings should be evaluated in the light of several limitations. First, we recognise the inherent limitations of register data, such as lack of confounder information, missing data or unknown data quality.[38] Second, information on comorbidities that might affect patient satisfaction was lacking. Third, the data were incomplete in 58% of the procedures. This is a major limitation of our study that affects the internal and external validity of our findings. Fourth, data on socioeconomic factors were lacking. The study of Iderberg et al[11] demonstrated that socioeconomic indicators were associated with outcomes of surgery for LSS.

## CONCLUSIONS

Patients surgically treated for the common lumbar spine diseases LSS or LDH, with negative future general health expectations, were significantly less satisfied with treatment compared with patients with positive expectations on future general health. The findings of this study can be used in the shared decision-making process when surgery is a treatment option for patients with LSS or LDH to establish realistic expectations and to enable patients to actively engage in rehabilitation.

**Contributors** All authors designed the study. AJ analysed the data. All authors interpreted the data. AJ wrote the manuscript with contributions from LS, PW, FGS and JK. All authors approved the final version of the manuscript. AJ is the guarantor.

**Funding** The authors have not declared a specific grant for this research from any funding agency in the public, commercial or not-for-profit sectors.

**Competing interests** None declared.

**Patient and public involvement** Patients and/or the public were not involved in the design, or conduct, or reporting, or dissemination plans of this research.

**Patient consent for publication** Not required.

**Ethics approval** This study involves human participants and was approved by the Swedish Ethical Review Authority (registration number: 2020-03557). Participants gave informed consent to participate in the study before taking part.

**Provenance and peer review** Not commissioned; externally peer reviewed.

**Data availability statement** Data may be obtained from a third party and are not publicly available. Data are available from the national Swedish spine register (Swespine) after approval by the Swedish Ethical Review Authority and according to the regulations in the General Data Protection Regulation and the Swedish Patient Data Act.

**ORCID iD**
Anders Joelson http://orcid.org/0000-0002-7931-9617

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
