## [Reviewer comments · BMJ Open]

ARTICLE DETAILS

TITLE (PROVISIONAL)	Associations between future health expectations and patient satisfaction after lumbar spine surgery: A longitudinal observational study of 9929 lumbar spine surgery procedures
AUTHORS	Joelson, Anders; Szigethy, Lilla; Wildeman, Peter; Sigmundsson, Freyr Gauti; Karlsson, Jan

VERSION 1 – REVIEW

REVIEWER	Jon D. Lurie Dartmouth College Geisel School of Medicine
REVIEW RETURNED	04-May-2023

GENERAL COMMENTS	This paper explores the interesting question of how pre-operative expectations of future health state affect outcomes of common lumbar spine surgeries. Unfortunately there are severe limitations in the data and the analysis that limit any insights or conclusions that can be withdrawn. While the authors acknowledge these limitations they seem to minimize their impact. The major concern is that the question 11c, which might be best described as a health trajectory question, is being used as a direct measure of future health expectation, which it is not. Who has a higher expectation of their future health, someone who says it is definitely true that their current health is excellent and they don't know if they expect their health to get worse or not (rated as low future health expectation in the current study) or some who states that their future health is poor (or say it is definitely false that their current health is excellent) and that it is mostly false that they expect it to get worse (rated as high future health expectation in the current study). Adding to the ambiguity of the meaning of this variable is the ambiguity of whether respondents are factoring in their expectations of their upcoming surgery or not in their assessment; the authors assume they are as an explanation for why the health trajectory seems to worsen postoperatively, but it is worth pointing out that the context of this question is for General Health and this is the one domain of the SF-36 that appears unaffected, or at least least affected, by surgery as seen in the figures. Finally though the discussion of limitations states very briefly that there is some acknowledgement of missing data, data is missing from over half the sample which is a very major limitation. The analysis is somewhat confusing. There is no direct statistical testing to allow the reader how likely the differences in satisfaction between the two groups is likely based on chance alone. It is not clear what they mean by a "linear" logistic regression model. the interpretation of the OR in the model appear naive or else the methods are inadequately described: they say future expectation has a major affect in the LSS group with an OR of 1.71 (which is a fairly modest affect) while age has only a small affect with an OR of 1.38 and they mention no affect of BMI with an OR of 1.25 without taking account
---

	that future health expectation has only 2 levels while the OR for age and BMI are presumably per unit if they were entered as continuous variables. They also provide no details , such as the C statistic, of whether the model meaningfully predicts the outcome at all. Again while they (very) briefly acknowledge the limited number of covariates, having only age gender and BMI to compare across groups greatly limits the ability to interpret or control for confounding. Finally one would assume that when they state future health expectation in the logistic model they are referring to the per-operative value of the health trajectory question, but since they have the same measure post-operatively it is important to clarify since just saying future health expectation could mean either one.
--	---

REVIEWER	Hanno S. Meyer University Hospital Hamburg-Eppendorf, Department of Neurosurgery
REVIEW RETURNED	12-May-2023

GENERAL COMMENTS	Dear colleagues: I would like to commend you on a concise manuscript. You present a study that is methodologically thorough and yielded clear results. There is one issue I would like you to consider: given that there is ample research on the relation between preoperative patient expectations and postoperative outcome measures, including patient satisfaction (such as the two published studies on the matter I was involved in and those you cite in your manuscript), why did we need another study? You mention that this was the first study ever to do this, but what is the rationale for / assumed benefit of using expectations on general health as opposed to expectations on specific disease-related outcomes? What was the difference between the two types of expectations that you had in mind when you initiated the study, and could you confirm this? I think the manuscript would benefit if this could be carved out more clearly in the Introduction and Discussion in order to avoid making it appear to be "just another expectation / PROM study in spine surgery".
---

VERSION 1 – AUTHOR RESPONSE

Reviewer 1:

Dr. Jon D. Lurie, Dartmouth College Geisel School of Medicine

Comments to the Author:

1. This paper explores the interesting question of how pre-operative expectations of future health state affect outcomes of common lumbar spine surgeries. Unfortunately there are severe limitations in the data and the analysis that limit any insights or conclusions that can be withdrawn. While the authors acknowledge these limitations they seem to minimize their impact. The major concern is that the question 11c, which might be best described as a health trajectory question, is being used as a direct measure of future health expectation, which it is not. Who has a higher expectation of their

future health, someone who says it is definitely true that their current health is excellent and they don't know if they expect their health to get worse or not (rated as low future health expectation in the current study) or some who states that their future health is poor (or say it is definitely false that their current health is excellent) and that it is mostly false that they expect it to get worse (rated as high future health expectation in the current study). Adding to the ambiguity of the meaning of this variable is the ambiguity of whether respondents are factoring in their expectations of their upcoming surgery or not in their assessment; the authors assume they are as an explanation for why the health trajectory seems to worsen postoperatively, but it is worth pointing out that the context of this question is for General Health and this is the one domain of the SF-36 that appears unaffected, or at least affected, by surgery as seen in the figures.

Answer: We agree with the reviewer that there are several limitations in using item 11c. We also agree that baseline health (e.g. excellent present health) might affect the interpretation of future health expectations. In our paper, we grouped the answers of item 11c into high and low health expectations. We used the words high/low but we could have used other wordings e.g. positive/negative, optimistic/pessimistic etc. We finally decided to use high and low. However, we agree with the reviewer that using high/low may add ambiguity. The example provided by the reviewer is very illustrative, a person who rates the current health as excellent and expects the future health to get somewhat worse may still have high health expectations. Looking at it now, it seems like positive/negative health expectations is a better alternative. In the revised manuscript we have replaced high/low with positive/negative throughout the manuscript. We have also clarified that items 1 and 11a-d form the GH domain of SF-36.

2. Finally though the discussion of limitations states very briefly that there is some acknowledgment of missing data, data is missing from over half the sample which is a very major limitation.

Answer: We agree that the missing data is a major limitation of our study and we do not want to minimize the importance of the limitations of our study. In the revised manuscript we have added that a sentence in the key points and in the limitations section stating that the missing data is a major limitation of our study and that this limitation affect the internal and external validity of our findings.

3. The analysis is somewhat confusing. There is no direct statistical testing to allow the reader how likely the differences in satisfaction between the two groups is likely based on chance alone.

Answer: We agree with the reviewer that this needs to be clarified. We have avoided p-values throughout the manuscript in favor of confidence intervals (CIs). In Table 2, the CIs are non-overlapping indicating that the results are statistically significant. In the revised manuscript we have added information about the non-overlapping CIs.

4. It is not clear what they mean by a "linear" logistic regression model. the interpretation of the OR in the model appear naive or else the methods are inadequately described: they say future expectation has a major affect in the LSS group with an OR of 1.71 (which is a fairly modest affect) while age has only a small affect with an OR of 1.38 and they mention no affect of BMI with an OR of 1.25 without taking account that future health expectation has only 2 levels while the OR for age and BMI are presumably per unit if they were entered as continuous variables.

Answer: We agree with the reviewer that the handling of age and BMI in our model needs clarification. Age and BMI were not entered as continuous variables. All covariates of the model were binary, continuous covariates were dichotomised by using their respective median values. This means that age and BMI only have 2 levels in our model. Moreover, we have de-emphasized the findings of the regression analysis throughout the paper, e.g. removed from the abstract. In the discussion we stress that the OR of future health expectations and present health perceptions had overlapping CIs indicating that there was no statistically significant difference. Concerning the use of "linear", we use the terminology suggested by Altman (ref 26) i.e. multiple linear logistic regression (e.g. to distinguish from multiple quadratic logistic regression).

5. They also provide no details , such as the C statistic, of whether the model meaningfully predicts the outcome at all.

Answer: The authors thank the reviewer for pointing this out. In the revised manuscript Table 4 we have added information on Hosmer-Lemeshow goodness-of-fit.

6. Again while they (very) briefly acknowledge the limited number of covariates, having only age gender and BMI to compare across groups greatly limits the ability to interpret or control for confounding.

Answer: We agree that our control of confounders is limited. The reason is that Swespine has very limited information about confounders. The most reliable information is on age, gender, and BMI so we decided to control only for these factors.

7. Finally one would assume that when they state future health expectation in the logistic model they are referring to the per-operative value of the health trajectory question, but since they have the same measure post-operatively it is important to clarify since just saying future health expectation could mean either one.

Answer: Thank you for pointing this out. Yes, we use the preoperative values for future health expectation and present health perceptions in our model. This is clarified in the revised manuscript.

Thank you for your most valuable comments.

Reviewer 2:

Dr. Hanno S. Meyer, University Hospital Hamburg-Eppendorf

Comments to the Author:

Dear colleagues:

I would like to commend you on a concise manuscript. You present a study that is methodologically thorough and yielded clear results.

1. There is one issue I would like you to consider: given that there is ample research on the relation between preoperative patient expectations and postoperative outcome measures, including patient satisfaction (such as the two published studies on the matter I was involved in and those you cite in your manuscript), why did we need another study? You mention that this was the first study ever to do this, but what is the rationale for / assumed benefit of using expectations on general health as opposed to expectations on specific disease-related outcomes? What was the difference between the two types of expectations that you had in mind when you initiated the study, and could you confirm this? I think the manuscript would benefit if this could be carved out more clearly in the Introduction and Discussion in order to avoid making it appear to be "just another expectation / PROM study in spine surgery".

Answer: The authors thank the reviewer for pointing this out. We certainly not want our paper to appear to be just another expectation study in spine surgery. In the revised manuscript we have

changed the introduction and we have added 2 references. We have also changed the first section of the discussion.

Thank you for your most valuable comments.

VERSION 2 – REVIEW

REVIEWER	Hanno S. Meyer University Hospital Hamburg-Eppendorf, Department of Neurosurgery
REVIEW RETURNED	22-Jun-2023
GENERAL COMMENTS	The authors have addressed my concerns, and the paper benefitted substantially from the changes they have made. I support publication in BMJ Open.